# LIMITATIONS OF PIECEWISE LINEARITY FOR EFFICIENT ROBUSTNESS CERTIFICATION

## ABSTRACT

Certified defenses against small-norm adversarial examples have received growing attention in recent years; though certified accuracies of state-of-the-art methods remain far below their non-robust counterparts, despite the fact that benchmark datasets have been shown to be well-separated at far larger radii than the literature generally attempts to certify. In this work, we offer insights that identify potential factors in this performance gap. Specifically, our analysis reveals that piecewise linearity imposes fundamental limitations on the tightness of leading certification techniques. These limitations are felt in practical terms as a greater need for capacity in models hoped to be certified efficiently. Moreover, this is *in addition* to the capacity necessary to learn a robust boundary, studied in prior work. However, we argue that addressing the limitations of piecewise linearity through scaling up model capacity may give rise to potential difficulties—particularly regarding robust generalization—therefore, we conclude by suggesting that developing *smooth* activation functions may be the way forward for advancing the performance of certified neural networks.

## 1 INTRODUCTION

Since the discovery of *adversarial examples* (Szegedy et al., 2014), defenses against malicious input perturbations to deep learning systems have received notable attention. While many early-proposed defenses—such as *adversarial training* (Madry et al., 2018)—are heuristic in nature, a growing body of work seeking *provable* defenses has arisen (Cohen et al., 2019; Croce et al., 2019; Fromherz et al., 2021; Huang et al., 2021; Jordan et al., 2019; Lee et al., 2020; Leino & Fredrikson, 2021; Leino et al., 2021; Li et al., 2019; Singla et al., 2022; Trockman & Kolter, 2021; Wong et al., 2018; Zhang et al., 2018). Generally, such defenses attempt to provide a certificate of *local robustness* (given formally in Definition 1), which guarantees a network's prediction on a given point is stable under small perturbations (typically in Euclidean or sometimes $\ell_\infty$ space); this precludes the possibility of *small-norm* adversarial examples on certified points.

The success of a certified defense is typically measured empirically using *verified robust accuracy* (VRA), which reflects the fraction of points that are both (i) classified correctly and (ii) certified as locally robust. Despite the fact that perfect robust classification (i.e., 100% VRA) is known to be possible on standard datasets at the adversarial perturbation budgets used in the literature (Yang et al., 2020b), this possibility is far from realized in the current state of the art. For example, on the benchmark dataset CIFAR-10, state-of-the-art methods offering deterministic guarantees of $\ell_2$ robustness[1] have remained at approximately 60% VRA (Huang et al., 2021; Leino et al., 2021; Singla et al., 2022; Trockman & Kolter, 2021), while non-robust models handily eclipse 95% accuracy.

It is difficult to precisely account for this discrepancy; though among other reasons, state-of-the-art methods typically use loose bounds to perform certification—as exact certification is (for general ReLU networks) NP-complete (Katz et al., 2017; Sinha et al., 2018)—which conceivably leads to falsely flagging truly robust points or to over-regularization of the learned model. While conservative approximations may be necessary to perform efficient certification (and to facilitate efficient robust training), it is certainly possible that they foil reasonable hopes for "optimality." In this work, we

---

[1]In this work we primarily consider certified defenses that provide a deterministic guarantee of local robustness, as opposed to a statistical guarantee. For further discussion of this point, see Section 4.

offer further insight into the shortcomings of modern certification techniques by analyzing their limitations in the context of the architectural settings in which they are conventionally employed.

In particular, we find that *piecewise linearity*—a practically ubiquitous property of neural networks considered in the certification literature (e.g., standard ReLU and the more recently popularized "MinMax" (Anil et al., 2019) activations are both piecewise linear)—fundamentally limits the power of *Lipschitz-based $\ell_2$* local robustness certification. In effect, we argue, this means that extra capacity is needed *simply for facilitating efficient certification*—in addition to whatever capacity may be required for learning a robust boundary (e.g., as examined by Bubeck & Sellke (2021)).

On the other hand, perhaps surprisingly, we prove that free from the constraint of piecewise linearity, Lipschitz-based certification is powerful enough to perform complete certification on any decision boundary, provided the implementation of the function giving rise to the boundary is under the learner's control (indeed, this is consistent with the fact that the highest performing certified defenses incorporate Lipschitz-based certification into training). These latter findings suggest that continued progress towards improving state-of-the-art VRA may be enabled through carefully chosen *smooth* activation functions,[2] which do not inherently limit the power of what are currently the most promising forms of efficient local robustness certification.

In summary, the primary contributions of this work are as follows: (1) we show that piecewise linearity imposes inherent limitations on the tightness of efficient robustness certification—our primary focus is Lipschitz-based certification, but we discuss similar limitations of other methods in Appendix B; (2) we prove that Lipschitz-based certification is fundamentally powerful for tight robustness certification, provided (i) the robust learning procedure has power over the implementation of the classifier, and (ii) the hypothesis class is *not* limited to piecewise linear networks; and (3) we demonstrate that tight Lipschitz-based certification may require significant capacity overhead in piecewise-linear networks. These findings offer a new perspective on the sticking points of modern certified training methods, and suggest possible paths forward.

We begin in Section 2 by introducing the limitations piecewise linearity imposes on robustness certification, starting generally, and narrowing our focus specifically to Lipschitz-based certification. We then discuss the role that capacity plays in mitigating these limitations in Section 3, which concludes with a discussion of the implications of our findings, both retrospectively and prescriptively. Finally, we discuss related work in Section 4, and offer our concluding remarks in Section 5.

## 2 LIMITATIONS OF PIECEWISE LINEARITY

The main insights in this work stem from the simple, yet crucial observation that the points lying at a fixed Euclidean distance from a piecewise-linear decision boundary, in general, do not themselves comprise a piecewise-linear surface. Therefore, in order for a certification procedure to precisely recover the set of robust points—those which lie a distance of at least $\epsilon$ from the decision boundary—it must be capable of producing a boundary between robust and non-robust points that is *not* piecewise-linear, even on networks that *are*. However, as we will see, Lipschitz-based certification, for example, is in fact constrained to produce a piecewise-linear "certified frontier" on piecewise-linear networks, as the set of just-certifiable points essentially corresponds to a level curve in the output of the network being certified.

On the other hand, if the level curves of the function being certified correspond (up to some constant factor) to their distance from the decision boundary (and must therefore include *smooth curves*), Lipschitz-based certification identifies precisely the points that are truly $\epsilon$-locally robust, provided a tight bound on the Lipschitz constant. As we will make clear, this has important implications regarding the power of Lipschitz-based certification in properly suited network architectures.

In the remainder of this section, we formalize this intuition and discuss some of its implications. Section 2.1 introduces our main theorem regarding the limitations imposed by piecewise linearity, along with the necessary background and definitions. Section 2.2 narrows the focus to Lipschitz-based certification, showing that despite being powerful in general, it is fundamentally limited within the hypothesis class of piecewise linear networks. Finally, Section 2.3 presents a thought experiment that provides basic intuition about the possible scale of the problems caused by these limitations.

---

[2]Or at least, activation functions which enable learning curved (as opposed to piecewise linear) functions.

## 2.1 FUNDAMENTAL LIMITATIONS TO CERTIFICATION COMPLETENESS

For our purposes, we will consider a neural network to be a function $f : \mathbb{R}^n \to \mathbb{R}^m$ mapping $n$-dimensional inputs to *logit* values corresponding to $m$ different classes. From the network function $f$, we derive a neural classifier, $F : \mathbb{R}^n \to \mathbb{R}^m$, by letting $F(x) = \operatorname{argmax}_{i \in [m]} f_i(x)$. When it is clear from the context which we are referring to, we will use the term "neural network" for both the network function $f$ and its corresponding classifier $F$. Note that two different neural network functions, $f$ and $f'$, may lead to the same predictions everywhere, i.e., $\forall x . F(x) = F'(x)$. When this happens, we say that $f$ and $f'$ share the same *decision boundary*, where the decision boundary is simply the set of points where $f_i(x) = f_j(x)$ for some $i \neq j \in [m]$.

In this work, we consider the problem of local robustness certification. As in prior work, we define local robustness as a property of a point $x$ and classifier $F$, parameterized by a *perturbation budget*, or *robustness radius*, $\epsilon$, as in Definition 1.

**Definition 1** ($\epsilon$-Local Robustness). *A classifier $F : \mathbb{R}^n \to [m]$ is $\epsilon$-locally robust at point $x \in \mathbb{R}^n$, with respect to norm $|| \cdot ||$, if*

$$\forall x' \in \mathbb{R}^n . \, ||x - x'|| \leq \epsilon \implies F(x) = F(x').$$

A *certification procedure*, cert, is a function that takes a neural network, $f$, a point, $x$, and a perturbation budget, $\epsilon$, and produces a label in $\{0, 1\}$, where an output of 1 means that $f$ is certified as $\epsilon$-locally robust at $x$. A valid certification procedure must be *sound*, i.e., $\text{cert}(f, x, \epsilon) = 1 \implies F$ is $\epsilon$-locally robust at $x$; however, it need not be *complete*, i.e., it may be the case that $\text{cert}(f, x, \epsilon) = 0$ and yet $F$ is in fact $\epsilon$-locally robust at $x$.

For a given certification procedure, let the *certified regions* of $f$, $C_{\text{cert}}(f, \epsilon) = \{x : \text{cert}(f, x, \epsilon)\}$ be the set of points that can be positively certified by cert. Similarly, let the *robust regions* of $f$ be given by the set $R(F, \epsilon) = \{x : F \text{ is } \epsilon\text{-locally robust at } x\}$ of $\epsilon$-locally robust points (note that, in contrast to $C_{\text{cert}}$, $R$ does not depend on the implementation of $f$, only its classification outputs, given by $F$).

Soundness entails that $\forall f . \, C_{\text{cert}}(f, \epsilon) \subseteq R(F, \epsilon)$, but clearly it is desirable for $C_{\text{cert}}(f, \epsilon)$ to match $R(F, \epsilon)$ as tightly as possible; when this is achieved perfectly we can consider cert to be "complete." However, as $C_{\text{cert}}(f, \epsilon)$ can depend on the underlying function, $f$, which has a surjective mapping to classifiers, $F$, derived from the same hypothesis class, we must be careful in defining completeness precisely. Let $\mathcal{F}$ be a *hypothesis class*—a family of functions of type $\mathbb{R}^n \to \mathbb{R}^m$, e.g., that are captured by some neural network architecture. We will also use the slight abuse of notation, $F \in \mathcal{F}$, to denote any $F : \mathbb{R}^n \to [m]$ such that there exists a function $f' \in \mathcal{F}$ which produces the same labels as $F$ on all inputs, i.e., $\forall x . F(x) = \operatorname{argmax}_{i \in [m]} f_i'(x)$. We say that a certification procedure, cert, is complete on $\mathcal{F}$ if all possible decision boundaries achievable by functions in the hypothesis class have at least one implementation in $\mathcal{F}$ for which cert perfectly recovers the true robust regions. This is stated formally in Definition 2.

**Definition 2.** *A certification procedure, cert, is complete on hypothesis class, $\mathcal{F}$, if for $\epsilon > 0$*

$$\forall F \in \mathcal{F} . \, \exists f' \in \mathcal{F} . \, \left( \forall x . F(x) = \operatorname*{argmax}_{i \in [m]} f_i'(x) \right) \wedge \left( C_{cert}(f', \epsilon) = R(F, \epsilon) \right)$$

Essentially, completeness over a hypothesis class entails a notion of compatibility between the certification procedure and the hypothesis class; specifically, it means that for any decision boundary expressible by the hypothesis class, it is possible for a learning procedure to produce a model that implements the decision boundary in a way that makes the certification procedure complete. Definition 2 provides a key relaxation from a stricter notion of completeness that would require $C_{\text{cert}}(f, \epsilon) = R(F, \epsilon)$ for all $f$, as this would not be achievable by any polynomial certification procedure[3] (Katz et al., 2017; Sinha et al., 2018). By requiring tight certification only modulo the decision boundary, we avoid this limitation, splitting the responsibility for completeness between the certification procedure, the learning algorithm, and the hypothesis class.

Next, we will also find it useful to define the *certified frontier* of $F$ under cert (Definition 3); essentially, the set of points that are just barely certified, which lie at the frontier of the certified

---

[3]Assuming $P \neq NP$.

regions. We will similarly define the *robust frontier* as the set of points that are just barely $\epsilon$-locally robust, which lie at the frontier of the robust regions.

**Definition 3** (Certified Frontier). *The certified frontier of a neural network, $F : \mathbb{R}^n \to [m]$, under certifier,* cert*, at perturbation budget, $\epsilon$, is the set of points*

$$\Delta\big(C_{cert}(f, \epsilon)\big) = \Big\{ x : \mathtt{cert}(f, x, \epsilon) \ \wedge \ \big(\forall \delta > 0 . \neg\mathtt{cert}\big(f, x, \epsilon + \delta\big)\big) \Big\}.$$

We now turn to the specifics of one of our main results, namely, that *piecewise linearity is a limiting factor for tight certification*. Of course, as alluded to earlier, some certification procedures *do* achieve complete certification on piecewise-linear networks—e.g., (Jordan et al., 2019; Tjeng et al., 2019)—however, such methods are invariably exponential. Thus, we characterize the set of *piecewise-linear limited* (PLL) methods in Definition 4. Intuitively, a certification procedure is PLL if it is constrained to produce piecewise-linear certified frontiers on piecewise-linear models.

**Definition 4** (Piecewise-linear Limited Certification). *A certification procedure,* cert*, is piecewise-linear limited (PLL) if*

$$\forall f . f \text{ is piecewise-linear} \implies \Delta\big(C_{cert}(f, \epsilon)\big) \text{ is piecewise-linear}$$

Note that the robust frontier of a network $F$ is, in general, *not* piecewise linear, even if $F$ (and thus its decision boundary) is piecewise linear. Thus, if the certified frontier of cert is piecewise linear, cert cannot be complete, i.e., $C \neq R$. Moreover, this means that any piecewise-linear limited certification procedure cannot be complete on the hypothesis class of piecewise linear networks (Theorem 1). The proof of Theorem 1 is given formally in Appendix A.1.

**Theorem 1.** *Any piecewise-linear limited certification procedure is incomplete on the hypothesis class of piecewise linear networks.*

The proof of Theorem 1 relies on the fact that a piecewise-linear function cannot be equal to a function exhibiting smooth curves. However, it is known that neural networks, *provided with enough capacity*, can approximate any function with arbitrary precision (Hornik, 1991). We address this point in Section 3, where we discuss the implications of Theorem 1 regarding the capacity requirements of tightly certifiable networks.

## 2.2 THE POWER AND LIMITATIONS OF LIPSCHITZ-BASED CERTIFICATION

We will now narrow our focus to consider the specific family of *Lipschitz-based* certification methods. Such methods perform certification by using an upper bound, $K$, on the network's Lipschitz constant; essentially, a point is certified if the margin by which the top-predicted class exceeds all other classes is greater than $\epsilon K$. In our work, we will set aside the details around how the Lipschitz is obtained, though this is also a source of potential looseness in the general approach. That is, we will (optimistically) take for granted that a tight bound is obtained in our analysis.

Lipschitz-based certification has proven effective in the literature, achieving state-of-the-art performance—when paired with an appropriate training routine—despite its simplicity (Leino et al., 2021; Trockman & Kolter, 2021). Lipschitz-based certification is advantageous in many ways; in addition to being easy to incorporate into a robust learning objective, it enables zero-cost certification at run time, as the Lipschitz constant does not need to be recomputed after training. On the other hand, it would seem that Lipschitz-based certification is fundamentally underpowered—the "global" Lipschitz constant is a conservative estimate of the local Lipschitz constant, which in turn gives a conservative estimate of how much the net output can change within a given neighborhood. If a primary sticking point for advancing certified accuracy is loose certification, it is fair to ask how promising Lipschitz-based certification will continue to be.

The philosophy behind incorporating Lipschitz-based certification into training is essentially that the potential shortcomings of Lipschitz-based certification can be addressed by learning a easily certifiable network function. We show that this intuition is essentially correct. Perhaps surprisingly, we show that Lipschitz-based certification is sufficiently powerful to be complete on the hypothesis class of Lipschitz functions[4] However, we also show that Lipschitz-based certification is PLL, meaning this potential cannot be achieved with a hypothesis class constrained by piecewise linearity.

---

[4]I.e., with bounded Lipschitz constant. Note that this is not a meaningful constraint for neural networks, as any neural network with Lipschitz activation functions and finite weights is Lipschitz in this sense.

### 2.2.1 LIPSCHITZ-BASED CERTIFICATION IS POWERFUL

We begin by showing that for any boundary achievable by a Lipschitz network function, when the learner is given control over the precise network function implementing the boundary, it is always possible to find an implementation that can be tightly certified using Lipschitz-based certification. This is stated formally in Theorem 2.

Theorem 2 further entails that there exists a network function for any $2\epsilon$-separated data that achieves perfect VRA under Lipschitz-based certification. The proof of Theorem 2 is given in Appendix A.2.

**Theorem 2.** *When the hypothesis class, $\mathcal{F}$, is given as the set of Lipschitz functions, Lipschitz-based certification is complete on $\mathcal{F}$.*

### 2.2.2 LIPSCHITZ-BASED CERTIFICATION IS LIMITED BY PIECEWISE-LINEARITY

Despite the power of Lipschitz-based certification for general functions, when restricted to the hypothesis class of piecewise linear networks, it becomes fundamentally limited. That is, formally, Lipschitz-based certification is PLL (Proposition 3).

**Proposition 3.** *Lipschitz-based certification is piecewise-linear limited.*

Proposition 3 follows essentially because the certified frontier of Lipschitz-based certification corresponds to a particular level curve of the network function, which is piecewise linear whenever the function is. As a direct consequence of Proposition 3 and Theorem 1, we arrive at Corollary 4.

**Corollary 4.** *Lipschitz-based certification is not complete on the hypothesis class of piece-wise linear networks.*

Note that taken in the context of Theorem 2, Corollary 4 means that in a sense, the fundamental limitation of Lipschitz-based certification is not intrinsic to its simplicity (e.g., because the local Lipschitz constant might be tighter than the global constant on some functions), but rather, it is related to the hypothesis class of networks being certified. Put differently, piecewise linearity imposes real limitations on Lipschitz-based certification that cannot be attributed to practical, but non-fundamental, issues, such as efficient computation of Lipschitz bounds, etc.

### 2.3 THE PROBLEM WITH CORNERS AND THE CURSE OF DIMENSIONALITY

The incongruence between the piecewise-linear certified frontier of Lipschitz-based methods, and the robust frontier of a piecewise-linear boundary, which features smooth curves, becomes relevant when the boundary comes to a "corner," or relatively sharp inflection point. At corners, the robust frontier curves at a fixed radius around the corner, while the certified frontier, absent aid from additional capacity (see Section 3), runs parallel to the facets forming the corner, offset by a fixed amount (see Figure 2 in Appendix D for an illustration). The sharper the corner, the larger the difference will be between the corresponding robust and certified regions. Additionally, we will see that this is also true the higher the dimension of the corner, i.e., the more independent half-spaces meet to create the corner.

As a thought experiment, we will model a $d$-dimensional corner as the intersection of $d$ orthogonal half-spaces. Assuming the level curves near the corner run parallel to the half-spaces, $h \in H$, forming the corner, in the best case, the certified region is given by the union of half-spaces obtained by flipping each $h \in H$ and shifting it by $\epsilon$. Consider the hypercube of width $\epsilon$ just opposite the corner. This hypercube lies entirely outside the robust region, meaning all points within it cannot be certified using Lipschitz-based certification. However, only the points intersecting the hypersphere of radius $\epsilon$ centered at the corner are truly non-$\epsilon$-robust. We can compute the ratio of the volume of the hypercube to the intersecting portion of the hypersphere, given by Equation 1:

$$\frac{\pi^{d/2}}{\Gamma\left(\frac{d}{2}+1\right)} \cdot \left(\frac{\epsilon}{2\epsilon}\right)^d \tag{1}$$

As the dimension increases, this ratio tends to zero, meaning that in high dimensions, *almost all points in this region opposite the corner are incorrectly uncertified*. Furthermore, the maximum distance from an uncertified point within this region to the boundary is equal to the diagonal of the hypercube, which is given by $\sqrt{d} \cdot \epsilon$. This means that even points that are *significantly more robust than required* may yet be uncertified.

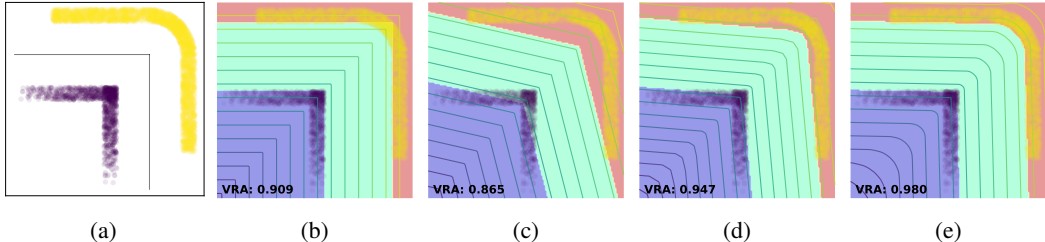

Figure 1: **(a)** synthetic dataset with two classes (shown in purple and yellow) derived to be positioned at distance $\epsilon$ from an ideal boundary that forms a 90° corner. **(b)** MinMax network implementing (with minimal capacity) an optimal boundary with respect to which all points are $\epsilon$-locally robust. The non-certifiable region is shown in green, and the level curves are also depicted. The Lipschitz constant of this network is 1, but not all points in the top right quadrant can be certified. **(c, d, e)** networks trained with $1\times$, $10\times$, and $100\times$ the minimal capacity required for the optimal boundary in (b). With $10\times$ capacity the learned VRA exceeds that of the minimal implementation, but near-perfect performance is not obtained with less than $100\times$ capacity. The code used to generate these examples is included in the supplementary material. Details are provided in Appendix C.

## 3 THE ROLE OF CAPACITY

The primary limitation of Lipschitz-based certification in piecewise-linear networks derives from the fact that we cannot have smoothly curved level curves in such networks (or, more generally, that PLL certification methods cannot have smoothly curved certified frontiers in such networks). However, while this is true in the strictest sense, a function with smooth curves can be approximated with arbitrary precision, given sufficient capacity. In other words, increased network capacity may be one possible option to mitigate the fundamental limitations discussed throughout Section 2. In this section, we investigate the capacity requirements necessary for tight PLL certification in piecewise-linear networks.

While the precise meaning of "capacity" in a quantifiable sense is a bit nebulous, for our purposes, we will consider capacity in a piecewise-linear network to correspond to the number of piecewise-linear regions. This grows with the number of internal neurons, though the relationship may vary depending on other aspects of the network architecture, e.g., the depth of the network.

Previous work has studied the capacity implications for learning a robust decision boundary, finding that separating points while controlling Lipschitzness may require additional capacity beyond what would be necessary to simply separate them (Bubeck & Sellke, 2021). Besides the capacity required to represent the decision boundary in a robust network, our work asks instead about the capacity required to *tightly certify* a given boundary. We find that in a piecewise linear network, even if the boundary is optimal—in that all points in the distribution are indeed a distance of $\epsilon$ or more from it—the network may require *additional capacity* to be able to prove this using the Lipschitz constant.

Taking the data distribution aside, we consider the goal of certifying all points that are sufficiently far from the boundary. As highlighted in Section 2.3, in places where the decision boundary forms high-dimensional "corners," there may be relatively large volumes of points that are $\epsilon$-far from the boundary but cannot be certified as long as the level curves simply run parallel to the boundary. In such cases, tight certification requires extra capacity specifically to round out the level curves around the corners in the decision boundary. We begin by demonstrating this concept via an illustrative example. We conclude by discussing the implications of our results and suggest avenues for future work.

### 3.1 AN ILLUSTRATIVE EXAMPLE OF HOW CAPACITY ENABLES TIGHT CERTIFICATION

As an example of how Lipschitz-based certification can require excess capacity beyond what is necessary to learn a robust boundary, we consider a synthetic 2-D dataset that can robustly separated by a simple piecewise linear boundary. An illustration is provided in Figure 1a. We begin with a decision boundary given by $B = \{(x_1, x_2) : \max(x_1, x_2) = 0\}$; this boundary separates points

with negative x- and y-coordinates from points in the other three quadrants, and forms a $90°$ corner at the origin. The data are then generated such that all the points with label 0 lie a distance of at least $\epsilon$ below and to the right of the boundary, and the points with label 1 lie a distance of at least $\epsilon$ above and to the right of the boundary. Specifically, the 1-labeled points curve around the boundary such that there is a tight margin of exactly $2\epsilon$ about the boundary.

By construction, the function $f(x) = [0, \max(x_1, x_2)]$ produces logit values that yield the boundary $B$, with respect to which all points in the dataset are $\epsilon$-locally robust. This function can be trivially implemented with minimal capacity by a simple MinMax network, $f(x) = \sigma(xW^1)W^2$, where $\sigma$ is the MinMax activation function, and $W^1$ and $W^2$ are given by Equation 2.

$$W^1 = \begin{bmatrix} 1 & 0 \\ 0 & 1 \end{bmatrix} \qquad W^2 = \begin{bmatrix} 0 & 0 \\ 0 & 1 \end{bmatrix} \tag{2}$$

Furthermore, the Lipschitz constant of $f$ is 1;[5] this can even be tightly obtained by taking the layer-wise product of the layer operator norms, as is typically done in practice. Hence, the points that can be certified will be those for which $|f_1(x) - f_0(x)| \geq \epsilon$; that is, the points outside the level curves $\max(x_1, x_2) = -\epsilon$ and $\max(x_1, x_2) = \epsilon$. However, we see that this certified frontier fails to certify many points in the positive x-y quadrant, despite the fact that all the points are indeed robust with respect to the boundary of $f$. This is depicted in Figure 1b.

In order to certify these points, we need the level curve corresponding to $f_1(x) - f_0(x) = \epsilon$ to bend smoothly around the boundary, rather than forming the same $90°$ angle. This requires more capacity.

To gain a sense of how this plays out in practice, we consider adding capacity via expanding the number of neurons in the hidden layer (which contained only two neurons in our minimal example). In Figures 1d and 1e, we show the boundaries of two additional learned networks, $g$ and $h$, with 20 and 200 internal neurons, respectively. We see that increasing the number of internal neurons by an order of magnitude yields a better set of level curves, but the network $g$ still must compromise as the level curves are not smooth enough to tightly follow the contour of the data. Finally, when we increase the number of internal neurons by *two orders of magnitude*, we at last obtain a function $h$ that achieves nearly 100% VRA on our sample data. This function, as desired, forms essentially smooth level curves that bend around the boundary corner with a radius of $\epsilon$. Interestingly, $h$ learns a boundary that is somewhat different from the boundary originally used to derive the data; however, both boundaries can be thought of as "equivalent" in the sense that they produce the same margin, reflecting that the optimal boundary for this dataset is not unique.

**Discussion.** In our example, we needed *100 times* more neurons than were necessary to construct an optimal decision boundary in order to tightly certify the boundary with the Lipschitz constant. While it is difficult to extrapolate from this toy example to a "real world" scenario, our results suggest that smoothing the level curves may require significant overhead beyond the capacity necessary to produce a truly robust boundary.

Another aspect of this experiment worth noting is that when the network had insufficient capacity to learn an optimally robust, tightly certified boundary (e.g., in Figures 1c and 1d), the resulting model tended to compromise by making the corner less sharp (compared to the desired $90°$ angle). Geometrically, when the boundary has an inflection with a wider angle, the difference between the certifiable frontier and the frontier of robust points is less pronounced (consider for example, what happens then the inflection approaches $180°$). In effect, this means that while under-parameterization of piecewise-linear models may be a problem for robust model performance in practice, this limitation may be (at least in part) manifested as an under-fit model as opposed to one with many robust but non-certifiable points. This is reflected in the empirical results for certifiably trained models in the literature, which typically have lower "clean accuracies" than their standard-trained counterparts. However, we note that these models also exhibit a discrepancy between their certified accuracy and their vulnerability to actual attacks, leaving the possibility that they may also fail to certify some truly robust points.

---

[5]More properly put, the Lipschitz constant of $|f_1 - f_0|$—which represents the margin by which the predicted class exceeds the non-predicted class—is 1.

## 3.2 POTENTIAL DRAWBACKS OF THE CAPACITY ESCAPE HATCH

As we have seen, by adding capacity, we can help overcome the limitations of piecewise linearity by enabling the network to approximate smooth curves around corners in the decision boundary. For universal tight certification, this needs to be done in the neighborhood of all corners on the decision boundary. To the extent that each corner requires independent capacity, hopes for the scalability of such an approach seem slim; albeit, VRA only requires tight certification on the data manifold, meaning that extra capacity should only be needed in places where the decision boundary has sharp inflections near in-distribution points.

However, this, too, presents an interesting problem. Namely, the network only has incentive to allocate capacity to round the level curves in the places that are necessary to certify its *training set*; i.e., where inflections in the decision boundary encroach on training points. Meanwhile, if similar inflections exist near test points not seen during training, the learned network may fail to certify them—even if the boundary is general, and even if it is also robust. In other words, we are faced with not only the challenge of learning a generally robust boundary, but additionally of learning a generally certifiable function. Indeed, generalization of VRA is empirically observed to be worse than the corresponding "clean accuracy" would indicate—a principle that has been noted in prior work due to its privacy implications (Yeom et al., 2020).

**A Proposed Way Forward.** Another possibility for addressing the fact that Lipschitz-based certification is PLL is to expand the hypothesis class to enable smooth curves in the decision surface. Ultimately, our analysis shows that Lipschitz-based certification is most effective when the level curves of the network function accurately reflect the $\ell_2$ distance to the boundary, which requires the possibility of smooth curves. This goal may be best achieved by purpose-built activations, as piecewise linearity stems from the choice in activation function.

State-of-the-art Lipschitz-based certifiable training methods have enjoyed increased success in recent years through leveraging MinMax activations (Anil et al., 2019)—or a variant thereof proposed by Singla et al. (2022)—which are piecewise linear. MinMax has a distinct advantage over the more common ReLU activation, due to its *gradient-norm-preserving* (GNP) property, which Anil et al. demonstrate is key for tight, efficient Lipschitz bounds. While the need for gradient norm preservation remains clear, we posit that some form of smoothness is an additional desirable property, as it would free the hypothesis class from piecewise linearity. We believe the task of designing suitable smooth activation functions for PLL-certified networks is a promising avenue for future work.

## 4 RELATED WORK

**Power and Limitations of Lipschitz-based Certification.** Several of the early efforts around robustness certification focused on post hoc certification of networks trained outside of the control of the certifier. This is a fundamentally hard problem, shown to be NP-complete by Katz et al. (2017) and Sinha et al. (2018). While this fundamentally limits the tractability of complete post hoc certification, the limitation is of lesser concern for modern approaches that incorporate certification into the training objective, thus encouraging learning models that better facilitate efficient certification.

The specific limitations of Lipschitz-based certification have also been of great interest in the prior literature. Most of these results particularly consider the practical problem of bounding a neural network's Lipschitz constant. For example, Huster et al. (2018) note that the common method of using the product of the layer-wise operator norm cannot tightly bound the Lipschitz constant of even basic functions in ReLU networks. Anil et al. (2019) study this point further demonstrating a trade-off between expressive power and efficient Lipschitz bound computation in networks with non-gradient-norm-preserving activation functions. This limitation is handled by using network architectures with gradient-norm-preserving activation function such as MinMax, and orthonormal linear operators (though the latter need not necessarily be strictly enforced as it is a learnable objective). Anil et al. conjecture that such networks are *universal 1-Lipschitz function approximators*, suggesting that learning any Lipschitz function in such a way that the Lipschitz constant can be bounded tightly and efficiently is possible. By contrast, our work points to previously unstudied limitations that are separate from the Lipschitz constant bounding problem, and are indeed not mitigated through the use of MinMax activations, which are piecewise linear. However, we propose that the limitations brought forth in our work may similarly be addressed via novel activation functions.

On the flip side, previous work has also touched on the power of Lipschitz-based certification. (Leino et al., 2021) showed that certification with the global Lipschitz constant can be as powerful as with the local Lipschitz constant when the model is under the learner's control. We extend this result in a number of key ways. First, we prove a stronger result that can be stated for all points, rather than for a finite set of points certified via the local Lipschitz constant. Second, we explicitly consider the hypothesis class, demonstrating that smoothness is a necessary condition to achieve this result.

**Capacity Requirements for Robust Neural Networks.** Understanding the role of capacity in deep neural networks has been a topic of interest in general, particularly due to the demonstrated effectiveness of highly over-parameterized models (Arora et al., 2018; Bubeck & Sellke, 2021; Du et al., 2019; Garg et al., 2022; Zhang et al., 2017). Recent work has also investigated this subject in the particular context of robust models. Bubeck & Sellke (2021) showed that under mild regularity assumptions, learning a highly accurate model with small Lipschitz constant requires significantly more parameters than would be required with no constraint on the Lipschitz constant—where the capacity overhead, in terms of the number of parameters, scales with the dimension. While a controlled Lipschitz constant is central to successful Lipschitz-based certification, our work (e.g., our example in Section 3.1), shows that a Lipschitz interpolation between points of opposite class is not sufficient for certification. As our analysis is focused on certification rather than Lipschitz interpolation, we complement the work of Bubeck & Sellke, showing that even further capacity may be required to appropriately bend the function's level curves to facilitate Lipschitz-based certification.

In addition to the information-theoretic capacity requirements, large numbers of parameters in deep networks may be necessary to facilitate efficient learning (Arora et al., 2018; Du et al., 2019). Recently, Garg et al. (2022) showed that robust learning in particular may require even greater over-parameterization than standard learning. Results such as these are complimentary to work such as ours, which focus on minimal parameterizations.

**Randomized Smoothing.** Our work has focused on deterministic certification. By contrast, *randomized smoothing* (Cohen et al., 2019; Lecuyer et al., 2018) has become a popular method that instead provides a *statistical guarantee* of robustness. Randomized smoothing (RS) essentially modifies the original function by predicting the expected label under Gaussian[6] noise. These predictions are empirically determined through sampling, with the statistical certificate depending on the unanimity of the sample labels. While RS provides a weaker robustness guarantee, it solidly outperforms deterministic methods in terms of certified accuracy. Interestingly, it seems clear that RS is *not* PLL, since it naturally smooths piecewise linear networks, leading to a smooth boundary and certified frontier—this may be one of the keys to its success. This observation gives further support to the notion that state-of-the-art deterministic methods may be held back by piecewise linearity, and may benefit from smooth activation functions.

## 5  CONCLUSIONS AND FUTURE DIRECTIONS

Incorporating Lipschitz-based certification into robust training procedures has proven to be the most effective way to achieve high deterministic $\ell_2$ verified-robust accuracy yet considered in the literature. Due to our Theorem 2, there is reason to believe Lipschitz-based certification has the power to remain as promising as current results suggest. However, we also showed that restricted to the hypothesis class of piecewise-linear networks, as has been the standard regime, Lipschitz-based certification becomes fundamentally limited. For piecewise-linear networks, this means that tight Lipschitz-based certification may require significantly more parameters, which, even if tractable, can complicate certifiably robust generalization (e.g., see Section 3.2). On the other hand, rather than viewing this as a fundamental drawback for Lipschitz-based certification, we propose that purpose-built activations—with the correct smoothness and gradient-norm-preserving properties—is a promising avenue for future work to free the most promising form of efficient deterministic certification from the limitations of piecewise linearity.

---

[6]Prior work has considered other distributions as well (Yang et al., 2020a)

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

# A   PROOFS

## A.1   PROOF OF THEOREM 1

**Theorem Statement.**   *Any piecewise-linear limited certification procedure is incomplete on the hypothesis class of piecewise linear networks.*

*Proof.* It suffices to show that there exists a boundary achievable by a piecewise-linear network for which no PLL certification method can tightly certify. We proceed by producing a piecewise linear boundary that induces a smooth robust frontier. This is sufficient to prove our theorem, as $\Delta\big(C_{\text{cert}}(f,\epsilon)\big) \neq \Delta\big(R(F,\epsilon)\big) \implies C_{\text{cert}}(f,\epsilon) \neq R(F,\epsilon)$.

Consider the 2-D boundary given by $\max(x,y) = 0$. Clearly, this boundary exists within the class of piecewise linear functions as the function $f(x,y) = \max(x,y)$ is piecewise linear. Now consider the points in the positive x-y quadrant. The points in this quadrant that are at distance $\epsilon$ from the boundary are given by $\sqrt{x^2 + y^2} = \epsilon$, which is not piecewise linear. By definition, any certification method that is PLL must have a certified frontier that is piecewise linear. Thus, the certified frontier of such any such method cannot be equal to $\sqrt{x^2 + y^2} = \epsilon$ in this quadrant. $\qquad\square$

## A.2   PROOF OF THEOREM 2

**Theorem Statement.**   *When the hypothesis class, $\mathcal{F}$, is given as the set of Lipschitz functions, Lipschitz-based certification is complete on $\mathcal{F}$.*

*Proof.* Let $\mathcal{F}$ be the set of Lipschitz functions. Consider the decision boundary of any function $f \in \mathcal{F}$. Define $f'$ as follows: let $d(x)$ be the minimum distance of $x$ from the decision boundary and let $f'(x) = d(x) \cdot \mathbb{1}_{F(x)}$, where $\mathbb{1}_{F(x)}$ is the one-hot encoding of $F(x)$.

First, observe that $f'_j - f'_i$ is 1-Lipschitz for all $i \neq j$. To see this consider the following. The Lipschitz constant is given by

$$\sup_{x,x'} \frac{\left|(f'_j(x) - f'_i(x)) - (f'_j(x') - f'_i(x'))\right|}{||x - x'||} = \sup_{x,x'} \frac{\left|f'_j(x) - f'_j(x') + f'_i(x') - f'_i(x)\right|}{||x - x'||} \qquad (3)$$

Consider points $x$ and $x'$, and let us assume that $||x - x'|| = \delta$. We would like to bound the quantity given by (4), the numerator in (3), by $\delta$.

$$\left|f'_j(x) - f'_j(x') + f'_i(x') - f'_i(x)\right| \qquad (4)$$

There are a few cases to consider. First if $F(x)$ and $F(x')$ are both different from $i$ and $j$, then (4) is $0 \leq \delta$. Since (4) is symmetric in both $i$ and $j$, and $x$ and $x'$, without loss of generality, we will assume $F(x) = j$. This leaves two cases: when $F(x') = j$, and when $F(x') \neq j$ (in the latter case we will not be concerned with whether or not $F(x') = i$).

In the first case we have

$$(4) = |f'_j(x) - f'_j(x')| = |d(x) - d(x')| \qquad (5)$$
$$= d(x) - d(x') \qquad \text{without loss of generality} \qquad (6)$$

Let $a$ be the nearest point on the boundary to $x'$, such that which $d(x') = ||x' - a||$. Thus,

$$d(x) \leq ||x - a|| \qquad \text{as } a \text{ is on the boundary} \qquad (7)$$
$$\leq ||x - x'|| + ||x' - a|| \qquad \text{by the triangle inequality} \qquad (8)$$
$$= \delta + d(x') \qquad (9)$$
$$\implies d(x) - d(x') \leq \delta \qquad \text{as desired} \qquad (10)$$

In the second case, $x$ and $x'$ are given different labels and we have

$$(4) = |f'_j(x) + f'_i(x')| \qquad (11)$$
$$\leq d(x) + d(x') \qquad \text{as } f'_i(x') \text{ is at most } d(x') \text{ (achieved when } F(x') = i) \qquad (12)$$

Since $x$ and $x'$ are given different labels, there must be at least one part of decision boundary that bisects the line segment connecting $x$ and $x'$; let $a$ be this intersection point. Additionally, since $a$ is on the boundary, we must have that $d(x) \leq ||x - a||$ and $d(x') \leq ||x' - a||$. Thus, as desired,

$$d(x) + d(x') \leq ||x - a|| + ||x' - a|| = \delta \tag{13}$$

This allows us to conclude that $f'_j - f'_i$ is 1-Lipschitz for all $i \neq j$, as claimed.

The points that are certified by Lipschitz-based certification are those for which (14) holds, where $j = F(x)$ and $K_{ji}$ is the Lipschitz constant of $f'_j - f'_i$.

$$\min_{i \neq j} \left\{ f'_j(x) - f'_i(x) - \epsilon K_{ji} \right\} \geq 0 \tag{14}$$

Notice that when $i \neq F(x)$, $f'_i(x) = 0$. Thus (14) can be simplified to $f'_j(x) = d(x) \geq \epsilon$, noting also that $K_{ji} = 1 \; \forall i, j$. Therefore, the points that can be certified via Lipschitz-based certification are those for which $d(x) \geq \epsilon$, which are precisely the points that are locally robust. $\square$

### A.3 PROOF OF PROPOSITION 3

**Theorem Statement.** *Lipschitz-based certification is piecewise-linear limited.*

*Proof.* Assume the function, $f$, being certified is piecewise linear. Without loss of generality, consider inputs $x$ for which the network predicts class $j$. The margin by which class $j$ surpasses all other classes is given by $m(x) = \min_i \{ f_j(x) - f_i(x) \}$. Note that $m$ is piecewise linear as $f$ is piecewise linear. Let $K$ be the Lipschitz constant of $m$. The largest radius that can be certified at $x$ is then $m/K$. Thus, the certified frontier is given by $m/K = \epsilon$; this corresponds to the level curve of $m$ corresponding to $m = \epsilon \cdot K$. Since $m$ is piecewise linear, this level curve is piecewise linear. Thus, the certified frontier is piecewise linear, and Lipschitz-based certification is PLL. $\square$

## B LIMITATIONS OF OTHER CERTIFICATION METHODS

### B.1 LIMITATIONS OF LOCAL-LIPSCHITZ-BASED CERTIFICATION

State-of-the-art deterministic $\ell_2$ certified performance is currently achieved using Lipschitz-based certification, which outperforms other types of certified training methods (Leino et al., 2021; Trockman & Kolter, 2021) such as those based on convex relaxations—e.g., (Wong et al., 2018)—or maximizing linear regions—e.g., (Croce et al., 2019; Xiao et al., 2019). Unsurprisingly, however, methods that use the *local* Lipschitz constant for certification can achieve similarly high VRA (Huang et al., 2021), though this comes at the cost of significantly slower certification.

The local Lipschitz constant at a point $x$ is given by $K_\epsilon(x)$ in Definition 5, which essentially corresponds to the maximum slope of the function within an $\epsilon$ neighborhood of $x$.

**Definition 5.** *The local Lipschitz constant is given by*

$$K_\epsilon(x) = \sup_{\substack{x_1, x_2 \, \cdot \, ||x - x_1|| \leq \epsilon \\ ||x - x_2|| \leq \epsilon}} \left\{ \frac{|f(x_1) - f(x_2)|}{||x_1 - x_2||} \right\}$$

*Local-Lipschitz-based certification*, similar to Lipschitz-based certification (Section 2.2), certifies points, $x$, when the margin by which the top-predicted class, $F(x)$, exceeds all other classes is greater than $\epsilon \cdot K_\epsilon(x)$.

While the local Lipschitz constant is always a lower bound for the global Lipschitz constant—and therefore local-Lipschitz-based certification can possibly be tighter—local-Lipschitz-based certification is nonetheless equally limited.

We will consider a generous setting in which the bound used for certification is exact, i.e., where the certification procedure has oracle access to $K_\epsilon(x)$. Because $K_\epsilon(x)$ is not piecewise linear, local-Lipschitz-based certification is not strictly piecewise-linear limited (PLL) in this setting. It is worth noting, however, that methods for approximating the local Lipschitz constant may not leverage this

smoothness in practice. Regardless, we show that local-Lipschitz-based certification is incomplete on piecewise-linear networks (Theorem 5).

This result is related to the fact that when the learner is given control over the implementation of the boundary, (global) Lipschitz-based certification can match the power of local-Lipschitz-based certification; this result has been proven in a slightly weaker formulation by Leino et al. (2021). We provide an alternative theorem statement and proof here that better aligns with the insights in this work.

**Theorem 5.** *Local-Lipschitz-based certification is not complete on the hypothesis class of piecewise-linear networks.*

*Proof.* It suffices to show that there exists a boundary achievable by a piecewise-linear network for which no corresponding piecewise-linear implementation can be tightly certified by local-Lipschitz-based certification. Recall that by Corollary 4 there exists such a boundary for (global) Lipschitz-based certification. We will consider one of the same such boundaries.

For a particular value of $\epsilon$, consider the points $\Delta\left(R(F, \epsilon)\right)$, which are at distance exactly $\epsilon$ from the boundary. There are two cases to consider: either (1) the local Lipschitz constant is always the same everywhere, i.e., $\forall \epsilon > 0, \forall x_1, x_2 \in \Delta\left(R(F, \epsilon)\right), K_\epsilon(x_1) = K_\epsilon(x_2)$, or (2) there is some variation in the local Lipschitz constant, such that $\exists \epsilon > 0, x_1, x_2 \in \Delta\left(R(F, \epsilon)\right)$ where $K_\epsilon(x_1) \neq K_\epsilon(x_2)$.

In the first case, we see that $K_\epsilon(x) = K$ (the global Lipschitz constant), meaning that local-Lipschitz-based certification will certify the exact same points as (global) Lipschitz-based certification. Thus, by Corollary 4, there must be a point which is robust at radius $\epsilon$ but not certifiable.

In the second case, without loss of generality, assume $K_\epsilon(x_1) > K_\epsilon(x_2)$. Because $f$ is piecewise linear, it is comprised of a finite number of linear functions, which in turn have a finite number of distinct slopes (gradient norms). Thus, if $K_\epsilon(x_1) > K_\epsilon(x_2)$, $K_\epsilon(x_1) - K_\epsilon(x_2) = \delta$ where $\delta$ belongs to some finite set of strictly positive values.

Furthermore, without loss of generality, $x_1$ and $x_2$ can be chosen to be arbitrarily close together, i.e., they lie arbitrarily near a point where the local Lipschitz constant changes. We will therefore consider $x_1$ and $x_2$ that are chosen such according to Equation 15.

$$\|x_1 - x_2\| < \frac{\epsilon \cdot \delta}{K} \tag{15}$$

Let $m_2$ be the margin by which the top-predicted class, $F(x_2)$, exceeds all other classes. The maximum radius that can be certified at $x_2$ is thus $m_2/K_\epsilon(x_2)$. Note that as certification is sound, we have

$$\frac{m_2}{K_\epsilon(x_2)} \leq \epsilon \tag{16}$$

Now consider the maximum radius that can be certified at $x_1$. Let $m_1$ be the margin by which the top-predicted class, $F(x_1)$, exceeds all other classes. The maximum radius that can be certified at $x_1$ is thus $m_1/K_\epsilon(x_1)$

$$
\begin{aligned}
\frac{m_1}{K_\epsilon(x_1)} &= \frac{m_1}{K_\epsilon(x_2) + \delta} && \text{by assumption} && (17)\\
&\leq \frac{m_2 + K\|x_1 - x_2\|}{K_\epsilon(x_2) + \delta} && \text{by definition of the Lipschitz constant} && (18)\\
&< \frac{m_2 + \epsilon \cdot \delta}{K_\epsilon(x_2) + \delta} && \text{by our choice of } \|x_1 - x_2\| \text{ in (15)} && (19)\\
&\leq \frac{\epsilon \cdot K_\epsilon(x_2) + \epsilon \cdot \delta}{K_\epsilon(x_2) + \delta} && \text{by (16)} && (20)\\
&= \epsilon && && (21)
\end{aligned}
$$

Thus, we see that $x_1$ cannot be certified with radius $\epsilon$, despite that its distance from the boundary is exactly $\epsilon$. $\qquad\square$

## B.2 Other Piecewise-linear Limited Methods

Our work focuses primarily on Lipschitz-based certification, which we demonstrate is fundamentally limited on the hypothesis class of piecewise linear networks. However, this limitation is not due specifically to the use of the Lipschitz constant per se; instead, we attribute it more generally to the fact that Lipschitz-based certification always produces a piecewise-linear certified frontier on piecewise-linear networks, a property we refer to as PLL (Definition 4). In this section we briefly discuss how this property may apply to other flavors of certification techniques that have been proposed in the literature.

**Convex Relaxations and Dual Networks.**   One classic approach for certification is through convex relaxation. A survey of such methods is given by Salman et al. (2019), who point out the limitations (regarding tight certification) of convex relaxations (though the authors do not consider our setting where the learner may control the implementation of the boundary, but rather focus on post hoc certification). Though many approaches in this family have been proposed, we will consider two baseline methods that capture a primal and dual formulation of convex relaxations: Fast-Lin (Weng et al., 2018), and an approach proposed by Wong & Kolter (2018), often referred to as "KW."

Fast-Lin directly derives upper and lower bounds on the output of a ReLU network in order to determine if an adversarial example might exist. This is done by iteratively computing upper and lower bounds for the neurons in each layer and using them to replace the ReLU activations with linear upper and lower bounds. This computation resembles a piecewise-linear network, suggesting that Fast-Lin is PLL.

The KW approach formulates the adversary as an LP that optimizes over the convex outer approximation of the set of top-level activations reachable through a norm-bounded perturbation. Crucially, for the sake of tractability, the LP can be bounded by the feasible set of the dual, which Wong & Kolter show can be expressed as a *dual network*, which resembles a backwards pass in the network being certified. For ReLU networks, the activations in the dual network are replaced with their upper convex envelopes (a linear function) over the bounded set $[\ell, u]$, where $\ell$ and $u$ represent lower and upper bounds on the pre-ReLU neural activations. The upper and lower bounds can be iteratively computed in a similar way to in Fast-Lip; thus, in its simplest form,[7] the dual network inherits the piecewise linearity of the original ReLU network being certified, suggesting the resulting certified frontier is piecewise linear, and certification is PLL.

**Hyperplane Projections.**   As exact certification is NP-complete, the literature has often turned to training procedures that help simple, approximate certification enjoy greater success. In piecewise linear networks, the input can be partitioned into a polyhedral complex where each convex region corresponds to a single *activation pattern*, over which the network is linear (Croce et al., 2019; Fromherz et al., 2021; Jordan et al., 2019). Motivated by this view of ReLU networks, one family of robust training approaches attempts to expand the linear regions of the network to simplify the combinatorial analysis of the possible ReLU activation patterns (Croce et al., 2019; Xiao et al., 2019). Croce et al. proposed a simple certification technique for networks trained with their "Maximum Margin Regularization" (MMR), where a point, $x$, is certified only if (1) the entire $\epsilon$-ball around $x$ is contained in a single convex activation region, and (2) the linear function corresponding to the region does not have a boundary within $\epsilon$ from $x$. This approach is clearly PLL, as the certified regions can be obtained by shrinking each activation region (possibly split in two if a linear decision boundary crosses it) by $\epsilon$. Since the original regions are convex polytopes, so too are the certified regions, thus the certified frontier is piecewise-linear.

In contrast to our findings for Lipschitz-based certification, it is worth noting that the limitations of this approach go beyond PLL, as completeness of the MMR approach is in direct conflict with non-linearity; and moreover, the approach is designed specifically for piecewise-linear networks.

## C Details on Experiments

The experiments presented in Figure 1 in Section 3 were performed using the `gloro` Python library, which implements the GloRo Net method of Leino et al. (2021) for training certifiably robust

---

[7]This approach has been refined in subsequent work that we do not consider here (Wong et al., 2018).

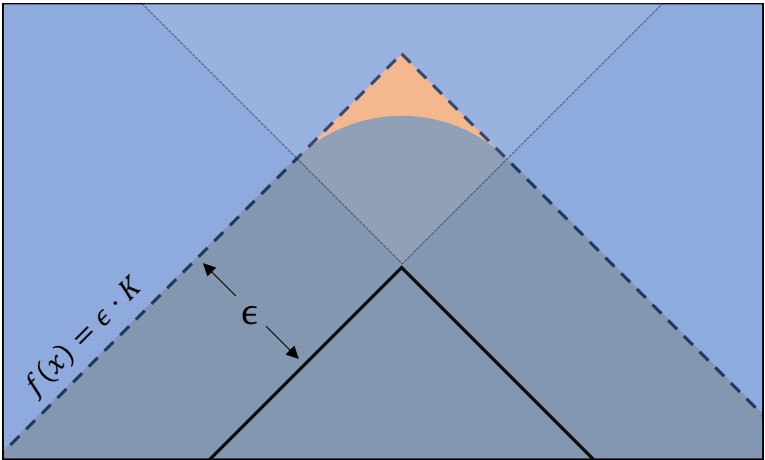

Figure 2: Illustration of the "corner problem" described in Section 2.3.

models by incorporating Lipschitz-based certification into training. All networks in the experiments consisted of a 1-hidden layer dense network with MinMax Anil et al. (2019) activations; three specific architectures were used, with 2, 20, and 200 hidden units, respectively. Models were trained for 64 epochs, with a batch size of 128. We chose hyperparameters inspired by those used by Leino et al. (see the original paper for details on the meaning of the various hyperparameters); namely, we used GloRo-TRADES loss with $\lambda = 1.2$, we scaled $\epsilon$ logarithmically to its ultimate value of $0.5$ by the half-way point of training, and we linearly decreased the learning rate from $10^{-3}$ to $0$ half-way through training.

## D  AN ILLUSTRATIVE EXAMPLE OF THE CORNER PROBLEM

For illustrative purposes a diagram is provided in Figure 2 that serves as a visual explanation of the "corner problem" described in Section 2.3. The boundary of a neural network, shown by the bold black line, forms a sharp corner. The complement to the robust region, i.e., the set of points that are *not* robust, is shown in gray. A simple implementation of this boundary has level curves that make similar sharp corners; the level curve corresponding to the certified frontier is shown by the dotted line, and the certified region is colored in blue. The region opposite the corner in the boundary is highlighted. We see that in this region, there is a set of points, shown in orange, that are *not* certified, despite the fact that they are robust, being at distance greater than $\epsilon$ from the boundary. In this two-dimensional example, these falsely flagged points make up a relatively small fraction of the uncertified points opposite the corner (represented as the union of the orange points and the highlighted gray points in the diagram); however, in high dimensions, virtually *all* uncertified points in this region would be falsely flagged, as indicated by Equation 1.

