# OpenReview forum: "Limitations of Piecewise Linearity for Efficient Robustness Certification"
_ICLR.cc/2023/Conference — Submitted to ICLR 2023_

### Official Review · Reviewer_gENj · 2022-10-24

**Confidence:** 4
**Correctness:** 4
**Technical Novelty And Significance:** 4
**Empirical Novelty And Significance:** Not applicable
**Recommendation:** 6

**Clarity, Quality, Novelty And Reproducibility:**

### Clarity

The paper does a good job explaining ideas, with clear definitions and no unnecessary jargon. I particularly liked the example in Section 3.1.

Some typos I noticed while reading the paper:

- Page 4: "Lipschitz-based certification has proven effective in the Literature"
- Page 6: "Would be necessary to simply separate them Bubeck & Sellke (2021)
- Page 9: "Due our Theorem 2"

### Quality
The paper advances our understanding of the role of capacity in Lipschitz-based _verifiability_. This builds on existing work that demonstrates that constraining the Lipschitz constant of the network (without concerning ourselves verifiability) means that additional capacity is required in the network. As outlined in Section 3.2, other researchers could build on this work to improve Lipschitz-based certifiable training methods by designing better hypothesis families.

### Novelty
To the best of my knowledge, the results and ideas in this paper are novel.

### Reproducibility
Reproducibility is not much of an issue as the paper is largely theoretical.

Minor issues:

- No details are provided on how the networks in Section 3.1 were trained. This makes it difficult to be certain that the order-of-magnitude increase in the number of neurons required to get to close to perfect VRA was not due to improper training. (For example: were the network trained adversarially? If so, what attack was used during adversarial training?)

**Strength And Weaknesses:**

The paper is clear and presents a novel idea that advances our understanding of Lipschitz-based verifiability.

The paper has two key weaknesses:

1. It is not clear what certification techniques are PLL; the paper could do a survey of leading techniques and characterize each of them. Even better: the paper could demonstrate that any non-PLL certification procedure must be exponential in the number of non-linear units.
2. The paper does not provide any hypothesis classes that are both gradient-norm-preserving and yet enable smooth curves in the decision surface. While showing how PLL certification procedures are limited is a strong result, the paper would have been made even stronger if paired with examples of hypothesis classes that reduce the excess capacity required. A negative result (i.e. no hypothesis class requires less capacity than piecewise-linear networks) would also be valuable.

Some suggestions:

- Suggestion: The title of the paper is very broad. What about specifying exactly when exactly limitations are encountered (e.g. "Limitations of Efficient Lipschitz-based Robustness Certification for Piecewise Linear Networks")
- Suggestion: Similarly, the abstract is vague on what certification techniques are affected by piecewise-linearity (it refers to "leading certification techniques", while only showing the Lipschitz-based methods are PLL). If no other certification techniques are shown to be PLL, the paper should either 1) provide evidence that Lipschitz-based methods significantly outstrip all other techniques 2) be more specific tha only Lipschitz-based methods are affected.

**Summary Of The Paper:**

The paper shows how piecewise-linear networks limit the tightness of "piecewise-linear limited" (PLL) certification procedures (such as Lipschitz-based certification). In particular, it demonstrates why piecewise-linear networks that require a small capacity to produce a robust boundary can require a much larger capacity to be _verifiable_ via a PLL certification procedure. Among the contributions of the paper include:

- Clear definitions for new ideas (e.g. completeness of certification procedures on a hypothesis class of decision boundaries; the notion of a certified frontier; PLL certification)
- Evidence (example + formal proof) explaining why PLL certification procedures can perform poorly around sharp inflection points in the boundary found in piecewise-linear networks.

**Summary Of The Review:**

This paper advanced our understanding of the limitations of Lipschitz-based verification techniques in verifying the robustness of piecewise-linear networks, and points to a way to reduce the excess capacity required for verifiability. As existing Lipschitz-based verification techniques are the state of the art for $l_2$ norm-bounded perturbations, this represents a valuable contribution to the field. Nevertheless, my enthusiasm for this paper is tempered by its failure to consider other verification techniques and the fact that it does not show that it is in fact possible to reduce the excess capacity currently required by piecewise-linear networks. (Otherwise, I would have rated it at 8, rather than the current 6). Keeping in mind that this is a conference paper of limited length, I would still vote to accept unless some other reviewer identifies an issue I missed.

---

> ### Author Response · Authors · 2022-11-11
> **Thank you for your review**
>
> Thank you for your review and feedback.
>
> > It is not clear what certification techniques are PLL; the paper could do a survey of leading techniques and characterize each of them.
>
> This is a good question. We introduced the notion of PLL to indicate that the reason Lipschitz-based certification is limited in the way it is does not have to do specifically with its use of the Lipschitz constant. We agree the notion would be more useful with a better sense of which methods have this limitation.
>
> Other candidates for PLL methods could include the method used by MMR training (Croce et al.), and methods based on convex relaxations (e.g., Kolter & Wong). We will update with more details on these methods.
>
> Local-Lipschitz-based methods do not seem to be PLL, but appear to be incomplete on the hypothesis class of piecewise-linear networks for different reasons. Specifically, in piecewise-linear networks, the local Lipschitz constant changes discontinuously (the local Lipschitz constant is the maximum slope in the piecewise linear regions contained in the neighborhood of the point be examined). Thus, the certified frontier jumps discontinuously between different (disjoint) level curves. Near the points of discontinuity, the certified radius cannot be tight, as the certified radius can change by a fixed amount with an arbitrarily small change in the input. We will include a formal argument of this sketch in the appendix to demonstrate that local-Lipschitz-based methods are also limited for certifying piecewise-linear networks (in addition to the fact that the true local Lipschitz constant may not be efficiently obtainable).
>
> In light of the observation above, fundamentally, the lack of completeness on the hypothesis class of piecewise-linear networks is the interesting property highlighted in this work. If PLL (specifically) is not a unifying cause, but yet if the phenomenon is widespread, we may de-emphasize the PLL concept, and simply include a survey of the limitations of other methods in the appendix.
>
> > Even better: the paper could demonstrate that any non-PLL certification procedure must be exponential in the number of non-linear units.
>
> This would indeed be an interesting result. Certainly, for example, complete methods (such as the branch & bound algorithms mentioned by reviewer `uK69`) are not PLL, but are also exponential. We believe the definitions presented in this work would enable subsequent work to consider such a hypothesis.
>
> > The paper does not provide any hypothesis classes that are both gradient-norm-preserving and yet enable smooth curves in the decision surface. While showing how PLL certification procedures are limited is a strong result, the paper would have been made even stronger if paired with examples of hypothesis classes that reduce the excess capacity required.
>
> Indeed, we believe this is a suitable endeavor for future work. The considerations involved would require both an in-depth analysis and a great deal of empirical validation, thus even with the answers to this question, the findings would likely be beyond the scope of what could be added to this paper.
>
> We think the ideas in this work give a good framework/perspective to design better hypothesis classes. However, the solutions are not entirely straightforward. First, "enabling smoothness" is sort of a minimum requirement to avoid piecewise linearity; however, the true goal is a hypothesis class over which Lipschitz-based certification can be complete, which may not be true of all smooth-enabling activations. Second, in the case of Lipschitz-based certification (as mentioned in the paper), gradient norm preservation is required to allow Lipschitz bounds to be accurately computed via efficient layer-wise products. This adds an extra constraint to consider. Finally, assuming there is some (non-empty) family of activation functions with the requisite properties, they would need to be evaluated empirically to determine if they can be used practically to achieve high performance in a neural network.
>
> Thus we believe it is appropriate to leave this question for future work.
>
> > A negative result (i.e. no hypothesis class requires less capacity than piecewise-linear networks) would also be valuable.
>
> This may be the case. If true, it would perhaps suggest that the way forward for Lipschitz-certifiable networks must be through ramping up capacity, e.g., by scaling to deep ResNet architectures. While this is possible in theory, it presents practical challenges, e.g., ensuring gradient norm preservation over skip/residual connections. This is also a promising direction for future work.

---

> > ### Author Response · Authors · 2022-11-11
> > **Response, continued**
> >
> > > If no other certification techniques are shown to be PLL, the paper should either 1) provide evidence that Lipschitz-based methods significantly outstrip all other techniques 2) be more specific that only Lipschitz-based methods are affected.
> >
> > As mentioned, we will investigate further whether other common methods are PLL and report back. While our emphasis is clearly Lipschitz-based methods, we will make this more clear if such methods are the only common PLL method in the literature.
> >
> > As for (1), we believe that the current evidence from the literature shows Lipschitz-based certification as the most promising method for $\ell_2$ certification. Lipschitz-based methods just exceed 60% VRA on standard benchmarks, while other methods, such as those based on convex relaxations (e.g., the method proposed by Kolter and Wong), or on maximizing linear regions (e.g., MMR, ReLU-Stability) score in the low 50s on the same tasks.
> >
> > The exception to this is _local_-Lipschitz-based certification, which unsurprisingly is competitive with (global) Lipschitz-based certification. However, we believe that local-Lipschitz-based methods are also limited in the piecewise-linear setting (we will update to confirm), and furthermore, local-Lipschitz-based certification is far more computationally expensive than using the global Lipschitz constant, while if the limitations described in this work are addressed, there may be no additional benefit to using a local Lipschitz constant.
> >
> > > No details are provided on how the networks in Section 3.1 were trained
> >
> > We included the code used to reproduce these experiments, but can also comment on this in the appendix. The training method used was GloRo training (Leino et al.), which results in networks that can be certified for robustness, and incorporates Lipschitz bounds during training to over-approximate a worst-case adversary. While training dynamics may also affect the capacity required, we believe that experimenting on actual Lipschitz-based methods gives a realistic idea of the capacity requirements that may be observed in practice.

---

> > > ### Comment · Reviewer_gENj · 2022-11-14
> > > **Response to author comment**
> > >
> > > > Other candidates for PLL methods could include the method used by MMR training (Croce et al.), and methods based on convex relaxations (e.g., Kolter & Wong). We will update with more details on these methods.
> > >
> > > Thank you - I'd be interested to see this.
> > >
> > > > We will include a formal argument of this sketch in the appendix to demonstrate that local-Lipschitz-based methods are also limited for certifying piecewise-linear networks (in addition to the fact that the true local Lipschitz constant may not be efficiently obtainable).
> > >
> > > This was not obvious to me, but makes sense once it's described as such. I'd appreciate seeing it in the Appendix!
> > >
> > > > (Regarding my suggestion to provide examples of hypothesis classes that reduce the excess capacity required): Indeed, we believe this is a suitable endeavor for future work. The considerations involved would require both an in-depth analysis and a great deal of empirical validation, thus even with the answers to this question, the findings would likely be beyond the scope of what could be added to this paper.
> > >
> > > Space shouldn't really be a limitation: the hypothesis class could be described in a paragraph of the main body, with a proof in the appendix showing that it reduces the excess capacity required / empirical results for some toy example. Having said that: I am sympathetic to the fact that it's difficult to find a hypothesis class reducing excess capacity during the course of the review period.
> > >
> > > > The exception to this is local-Lipschitz-based certification, which unsurprisingly is competitive with (global) Lipschitz-based certification. However, we believe that local-Lipschitz-based methods are also limited in the piecewise-linear setting (we will update to confirm)
> > >
> > > Thanks, I look forward to hearing your results.
> > >
> > > > We included the code used to reproduce these experiments, but can also comment on this in the appendix.
> > >
> > > I think it's helpful to provide details in the appendix - it's often hard when reading someone else's code to pick up on all the details of what they did (and details can often affect results significantly).

---

> > > > ### Author Response · Authors · 2022-11-18
> > > > **Updates in paper**
> > > >
> > > > Thank you for your reply. We have updated the paper with
> > > > * A detailed proof in Appendix B1 of how local-Lipschitz-based certification is also limited on piecewise linear networks. In a sense this boils down to the fact that the local Lipschitz constant is not fundamentally more powerful than the global. When the local Lipschitz constant is not uniform everywhere, it introduces looseness.
> > > > * A discussion of other PLL methods in Appendix B2. We considered Fast-Lin, KW, and MMR. Fast-Lin and KW are meant to be representative of primal and dual convex relaxation approaches, though they are perhaps the simplest iterations in a larger line of work. The precise analysis would depend on how upper and lower bounds are computed, but in their simplest form they can be computed using a piecewise linear "network," suggesting the method would be PLL. The method used by MMR is more clearly PLL, though its incompleteness is perhaps apparent even without this observation.
> > > > * Fixes to various typos/framing & comments from the reviews in general.

---

### Official Review · Reviewer_uK69 · 2022-10-25

**Confidence:** 5
**Correctness:** 2
**Technical Novelty And Significance:** 3
**Empirical Novelty And Significance:** 2
**Recommendation:** 5

**Clarity, Quality, Novelty And Reproducibility:**

The paper has moderate clarity. Claims are new but not well supported by evaluation results.

**Strength And Weaknesses:**

### Strength

The authors present an interesting theoretical discussion regarding Lipschitz-based certification with regard to piece-wise linear activations.

### Weakness
- Branch and bound (BaB) is widely adopted to break the barrier for piece-wise linear activations and enable complete certification. Representative works include the recent VNN-COMP winner alpha-beta-crown, as well as other participants like OVAL, VeriNet, Marabou, ERAN, etc (see VNN-COMP21 report [A] for details and VNN-COMP22 for updates). These SOTA complete verifiers significantly scale certification to larger models, speeding up certifications, and generalizing to different robustness properties and architectures. The paper does not mention any of them in related works or evaluation. Analysis and discussion on this are crucial to making claims solid and convincing.
- To support the power/limitations of Lipschitz-based certification approaches, it is necessary to use existing Lipschitz-based certification approaches to confirm the theoretical observations and insights on either toy examples or standard robustness benchmarks. Also, many existing works including some tools mentioned before support smooth activations. Experimental evaluation should be presented to support using smooth activations over piece-wise linear activations.

**Summary Of The Paper:**

This paper discusses the fundamental problem for certification using piece-wise linear activations like ReLU. Lipschitz-based certification is mainly investigated and advocated with smooth activations.

**Summary Of The Review:**

Overall, the paper missed extensive evaluations to support the proposed theoretical claims. Also, key components (branch and bound approaches) missing in theoretical analysis.

---

> ### Author Response · Authors · 2022-11-11
> **Thank you for your review**
>
> Thank you for taking the time to review this work. We disagree with the importance placed on complete methods, but will expand our discussion of the topic. Essentially, B&B methods achieve complete certification at the cost of exponential worst-case runtime. Our work discusses paths forward for _efficient_ methods. More on this below.
>
> ### Exact post hoc certification methods
>
> In the intro we briefly mention exact methods, to provide context on why we _don't_ focus on them in our work. We did not spend considerable time defending this choice, as our goal was to provide insights for improving approximate methods like Lipschitz-based certification, not to discourage work on exact post hoc certification, despite the fact that we are of the opinion that training procedures centered around efficient (but relaxed) certification are more promising for obtaining provably robust models. This point is mentioned again in the related work, but we will expand this discussion a bit (in Section 4) to make the following points more clear.
>
> Basically, exact certification methods---i.e., those that are complete on arbitrary networks, not merely in the sense of being "complete on a hypothesis class" as in Definition 2---are _not_ efficient. We mean this in the precise sense that the problem of exact post hoc certification has been shown to be NP-complete. Thus, some form of relaxation is necessary to enable "real-time" certification.
>
> What about these branch and bound methods then? They are perhaps more limited than Reviewer `uK69`'s comment would suggest. Let us consider the results reported in the VNN-COMP competition. First, this competition is entirely focused on $\ell_\infty$ certification rather than $\ell_2$ certification; the latter is our focus. While some methods, like CROWN, can handle Euclidean geometry, other methods from the competition do not even implement $\ell_2$ certification. Regardless, B&B techniques are clearly best suited to $\ell_\infty$ geometry, so it is unclear what the competition results entail for the $\ell_2$ case, and this was never tested.
>
> More importantly, while the results from the competition are impressive, the methods may be limited in how they can be used in certain applications because they are far from real-time. The time budget allocated to each instance in the competition is typically around _5 minutes_, which is orders of magnitude beyond the cost of running a NN forward pass. Even with this large budget, the methods frequently time out, meaning no certificate or counter example can be obtained. E.g., on the `cifar10-resnet` task in CNN-COMP21, alpha-beta-CROWN, the leader by a considerable margin, managed to decide only 58/72 of the given points in the allotted time. Thus, Despite the fact that the methods are complete, this is not especially meaningful if they fail to terminate in a reasonable timeframe.
>
> Note that by contrast, Lipschitz-based certification _is_ real-time. The extra cost of certification at runtime is essentially _zero_ since the Lipschitz constant can be computed once in advance. Moreover, state-of-the-art $\ell_2$ VRA is achieved by Lipschitz or local-Lipschitz-based methods. While VRA is less often the metric used to evaluate post hoc methods, we are not aware of any results that show VRA can be practically improved by certifying with these methods (since their completeness is counterbalanced by the possibility of timing out).
>
> Moreover, the validity of this work is not impacted by these methods. Exact methods are clearly not PLL, so the limitations discussed do not apply to them. On the other hand, this work is intended to provide insights that will help improve the VRA of efficient, "relaxed" certified training. Among such methods, Lipschitz-based certification is currently leading the pack for $\ell_2$ robustness. If the tightness of Lipschitz-based certification can be meaningfully improved, this would be a win over complete post hoc certification because of the massive performance benefits of relaxed certification.

---

> > ### Author Response · Authors · 2022-11-11
> > **On experiments**
> >
> > > To support the power/limitations of Lipschitz-based certification approaches, it is necessary to use existing Lipschitz-based certification approaches to confirm the theoretical observations and insights on either **toy examples** or standard robustness benchmarks.
> >
> > We _do_ provide examples of Lipschitz-based certification on toy examples to illustrate the insights (see Figure 1). On higher dimensional data it would, of course, be difficult to visualize in the same way. Even determining if points are truly robust, but residing by an uncertifiable "corner" of a high dimensional decision boundary would be prohibitively expensive on "standard" models.
> >
> > That is not to say the intuition drawn from our theoretical analysis is without outside evidence. For example, a gap between _empirical "robust" accuracy_ (ERA; obtained by attempting to find adversarial examples) and VRA is generally observed on state-of-the-art robust models. ERA can be seen as an upper bound on the robustness of the model, thus this gap leaves open the possibility that many points are robust, but just couldn't be certified. This is consistent with our theoretical insights. Furthermore, the inability to certify these "corner" points might instead lead underparameterized models to simply learn underfit, straighter boundaries (as observed in our experiments). This is consistent with the prior literature, which finds certifiably robust models to have lower "clean" accuracy. But this would complicate experimental validation of our insights, as it is an indirect effect. These points are made briefly in the "Discussion" section of 3.1.
> >
> > Finally, the validity of our theorems is not dependent on empirical validation. Their proofs are included in the appendix.
> >
> > > Many existing works including some tools mentioned before support smooth activations. Experimental evaluation should be presented to support using smooth activations over piece-wise linear activations.
> >
> > Indeed, Lipschitz-based certification easily accommodates 1-Lipschitz smooth activations (e.g., softplus). However, smoothness is sort of a minimum requirement; the true goal is a hypothesis class over which Lipschitz-based certification can be complete, which may not be true of all smooth-enabling activations. Moreover, in the case of Lipschitz-based certification (as mentioned in the paper), gradient norm preservation is required to allow Lipschitz bounds to be accurately computed via efficient layer-wise products. That is, without a proper smooth activation function, looseness may instead be observed due to looser Lipschitz bounds, even if the "corner problem" is solved. This adds an extra constraint to consider. Altogether we believe there are several challenges to designing suitable activation functions that would alleviate the limitations identified in this work; these challenges are more suitable for future work.

---

> > > ### Comment · Reviewer_uK69 · 2022-12-01
> > > **Thanks for the response**
> > >
> > > Thank the authors for the response and updated version of the paper incorporating other reviewers' comments. The arguments in response on BaB somewhat makes sense but should be supported with extensive experimental results. Overall, I increased my score to 5.

---

### Official Review · Reviewer_SHQ9 · 2022-10-25

**Confidence:** 3
**Correctness:** 3
**Technical Novelty And Significance:** 2
**Empirical Novelty And Significance:** Not applicable
**Recommendation:** 5

**Clarity, Quality, Novelty And Reproducibility:**

The writing is imprecise in parts and I hope the authors make edits to address the minor comments. The ideas outlined are novel and interesting. The authors have provided code to reproduce their experiments.

**Strength And Weaknesses:**

On a broad scope my take-away from the results of the paper feels very different to the author's inference and I am eager to hear the author's opinion on this and ready to change my mind on the following observations.

- In Theorem 2, why is the hypothesis class cal{F} restricted to be set of Lipschitz functions? Even for arbitrary label classifiers f, one can  construct f' based on d(x) : the distance to boundary. Additionally if one has access to d(x), then there is no certification to speak of? The label classifier f is epsilon-locally robust at x for all epsilon <= d(x) by definition..? So this result does not meaningfully show that Lipschitz-based certification is a good approach. In this ideal case, there is no need for certification. Additionally, even when "the learner is allowed flexibility over the precise flexibility over the network function" shouldn't f' also be a neural network of the same architecture? How is the f' constructed a valid instance?

- The fundamental issue appears to be a mismatch between level curves of a hypothesis and the shape of the data manifold. This issue is independent of Lipschitz-based certification right? The authors show in Figure 1 that if the data curved but the function class is piece-wise linear then one can find pockets that escape certification analysis. However if one uses this information to switch the function class to be smooth, couldn't we have the reverse issue? i.e. when the data is piece-wise linear opposite the boundary induced by the function which is now curved? Further, at different parts of the image manifold the local shape can vary in regularity and thus any structured hypothesis might necessarily need more parameters to learn a good boundary. If this thought is sound, then the take-away is that any certification is only as good as how well the regularity of the hypothesis matches shape of data manifold across the distribution.

## Minor Comments/Feedback
- The analysis implicitly assumes that the unit ball induced by the norm is a curved surface (which can reduce to piece-wise linear when the norm is \ell_1 for e.g.) so I request that the authors explicitly mention this upfront.
- I would appreciate if the authors remove overloaded notation and have a single use of F, f and cal{F} and of the term "network function". O
- In page 3,"Note that two different neural networks f, f' may lead to the same predictions everywhere", are there examples of such networks f, f' where the weights aren't equivalent modulo layer-wise scaling? Also in this statement can f,f' be different architectures?
- In Def 3, it should be  \forall delta>0, \neg cert(f, *x*, \epsilon + delta)
- Can the authors provide an example of when the robust frontier of a network is not piecewise linear? Is this only limited to situations where the data is supported on a curved (or non piecewise-linear) surface?
- In Appendix A.1, proof of Theorem 1, should it be \sqrt{x^2+y^2} = \epsilon ?
- In Appendix A.2, proof of Theorem 2, should it be "d(x) be the minimum distance of *x* from ..."
- I request that the authors add an explicit proof of Proposition 3 even if obvious.


**Summary Of The Paper:**

This paper attempt to understand the fundamental limitations of Lipschitz-based certification for piecewise-linear hypothesis classes. They show that piece-wise linear certification methods cannot sufficiently capture a true robust frontier of data that is curved and hence might need additional parameters. I commend the authors for proposing a unique viewpoint for understanding the gap between existing robustness certificates vs actual robust accuracy.

**Summary Of The Review:**

I think this paper undertakes an interesting perspective on why there is a gap between Lipschitz-based certificates and robust accuracy we observe in practice. However for reasons outlined above, I feel that the connection to Lipschitz certification is weak and this work while interesting needs to further evolve before it can be published.

---

> ### Author Response · Authors · 2022-11-07
> **Thank you for your review**
>
> Thank you for your review. We address your comments below, beginning with your first set of major observations, and continued as a reply to this thread.
>
> > Why is Thm 2 stated only for Lipschitz functions?
>
> Basically, only Lipschitz functions are useful for LC-based certification, and this assumption is mild, for the reasons stated in footnote 4. While the construction we give does not explicitly the Lipschitzness of the functions in the hypothesis class, the assumption allows us to avoid dealing with edge cases where the boundary corresponds to a non-Lipschitz function, e.g., $\sin(1/x)$, where the interpretation of the decision boundary is rather unclear.
>
> > Additionally if one has access to d(x), then there is no certification to speak of? The label classifier f is epsilon-locally robust at x for all epsilon <= d(x) by definition?
>
> If the network does indeed encode d(x), then certification does follow naturally. However, there are two issues. First, we don't know d(x) in advance, and in fact, d(x) references the boundary, which is also not known in advance. Instead we have to learn a function f, which we hope resembles the construction given in the proof of Theorem 2 (which uses d(x)). Then, given that we don't know d(x), or whether f even encodes it, we still need to perform certification on new points. Calculating d(x) is NP-complete on general networks (since it is equivalent to certification), so we cannot count on using d(x) directly for certification. Instead, we can use, e.g., Lipschitz-based certification, which can be run efficiently; and if f happens to encode d(x), then this certification will happen to be tight, even though we didn't know in advance that we were in this "ideal case."
>
> > Shouldn't f' also be a neural network of the same architecture? How is the f' constructed a valid instance?
>
> In our theorem, we assume f' is from the same _hypothesis class_ as the boundary being considered. The two hypothesis classes we consider are "all piecewise linear functions" or "all Lipschitz functions." It is fair to say that these hypothesis classes are a bit theoretical; with a fixed feed-forward architecture, the true hypothesis class is decidedly narrower. Regardless, we think the theoretical results are insightful, as they identify an issue with piecewise linearity, and hold "in the limit" as the hypothesis class approaches these two ideals. In practice, we would want completeness on the set of boundaries that are meaningful for data distributions we care about, which may be achievable with some fixed architecture, even if widening the hypothesis class to architectures that can achieve this opens the door to new boundaries that require an even wider hypothesis class to certify. The big takeaway is that the hypothesis class needs to be suitable to tightly certifying the boundaries you care about, which is impossible for piecewise linear networks.

---

> > ### Author Response · Authors · 2022-11-07
> > **Addressing minor comments**
> >
> > We will update the paper with the typos raised in Def 3 and Appendices A1 and A2, and will add a formalized version the proof sketch of Proposition 3 to the appendix.
> >
> > > The analysis implicitly assumes that the unit ball induced by the norm is a curved surface ... so I request that the authors explicitly mention this upfront.
> >
> > Correct, our work focuses on $\ell_2$ certification. We intended to make this explicit (from the intro: "we find that piecewise linearity... fundamentally limits the power of Lipschitz-based $\ell_2$ local robustness certification."), but can clarify if our focus was not apparent enough.
> >
> > > Are there examples of such networks f, f' where the weights aren't equivalent modulo layer-wise scaling? Also in this statement can f,f' be different architectures?
> >
> > Yes. We have illustrated some examples [here](https://ibb.co/jygS6rV). As one example, we can consider the instance used in the proof of Theorem 1 in Appendix A1. Let $f(x,y) = \max(x, y)$ and let $f'(x,y) = \sqrt{x^2 + y^2}$ if $x, y > 0$ and $f'(x,y) = \max(x, y)$ otherwise. The boundary at $f(x, y) = 0 = f'(x, y)$ is the same, but the functions are different. This can also be done even with networks that have the same architecture, as shown in our illustration linked above.
> >
> > > Can the authors provide an example of when the robust frontier of a network is not piecewise linear? Is this only limited to situations where the data is supported on a curved (or non piecewise-linear) surface?
> >
> > The proof of Theorem 1 in Appendix A1 gives a concrete example of this. Note that the certified frontier does _not_ have to do with the data, only with the boundary. The fact that the robust frontier of this boundary, $\max(x, y) = 0$, is not piecewise linear can be relevant even if the data is not on a "curved" surface. E.g., suppose we have two points, $p_1 = (-1, -1)$, and $p_2 = (\frac{1}{\sqrt{2}}, \frac{1}{\sqrt{2}})$. With just these two points, it doesn't really make sense to say the data are on a curved surface. Nonetheless, note that both points are robust w.r.t. the boundary, but $p_2$ cannot be certified when the boundary is implemented as $f(x,y) = max(x, y)$.

---

> > > ### Author Response · Authors · 2022-11-19
> > > **Updates in paper**
> > >
> > > We have updated the paper with fixes to the typos raised in this review, and a formal proof of Proposition 3 in the appendix, in addition to incorporating feedback from the other reviewers, such as a discussion of the limitations of other methods (besides Lipschitz-based certification) in similar settings. We hope you will find our response and updates helpful, and would be happy to incorporate more of the discussion in this thread to the paper if you believe it would enrich the final work.

---

> > ### Author Response · Authors · 2022-11-07
> > **Understanding the root of the issue identified**
> >
> > > The fundamental issue appears to be a mismatch between level curves of a hypothesis and the shape of the data manifold.
> >
> > The **issue we identify doesn't really have to do with the data manifold**, but rather, the boundary being certified. That is, the issue is a mismatch between the level curves of a hypothesis and _the certified frontier of boundaries in that hypothesis_. Of course, the boundaries we'd be most interested in certifying would be the ones that robustly separate some data distribution that we care about, but the theorems are more general than that. Essentially, whether certification is tight is a separate question from whether the network correctly labels its inputs or if points on the manifold are robust, although if the data are 2epsilon separable, then we could potentially achieve all at once.
> >
> > > This issue is independent of Lipschitz-based certification right?
> >
> > Yes and no. While we study Lipschitz-based methods in particular, the issue applies more broadly methods sharing the PLL property of Lipschitz-based certification. But as noted above, the issue does not have to do with the data manifold.
> >
> > > If one uses this information to switch the function class to be smooth, couldn't we have the reverse issue? i.e. when the data is piece-wise linear opposite the boundary induced by the function which is now curved?
> >
> > This is an interesting point that highlights the challenges of designing suitable activation functions, which we propose as a promising topic for future work. We agree that "smooth activation functions" will need to have many particular properties to address the issues raised in this work; fundamentally, the hypothesis class needs to be able to express functions which encode the distance to their own boundaries to resolve this limitation (in addition to other desirable properties, like gradient norm preservation, etc.). It is even possible that no such activation functions exist, in which case capacity may be the only way forward.
> >
> > But to answer this question more specifically, first, if the function class cannot have sharp corners, then the decision boundary cannot have sharp corners, and thus the problem with the opposite corner may not apply. Second, the function d(x) admits smooth curves, but _also_ sharp corners (on boundaries with sharp corners). We tried to be clear, but may have been informal at times, that we propose activations that _admit_ smooth curves, not ones that _preclude_ sharp corners. However, even precluding sharp corners may be acceptable if "strictly smooth" boundaries can be tightly certified.
> >
> > > Further, at different parts of the image manifold the local shape can vary in regularity and thus any structured hypothesis might necessarily need more parameters to learn a good boundary.
> >
> > This is actually a separate question, which is addressed in other work (e.g., Bubeck & Sellke). As mentioned above, whether a certifiable boundary correctly labels all instances is a separate question from whether a boundary can be tightly certified. Our work addresses the latter independently of the former. So here's where our work comes in: suppose you learned a boundary that does epsilon-separate the data (using however much capacity is necessary for the given data). If that boundary is piecewise linear then there will be pockets of points that are robust but not certifiable. If you want to fill them in, you will need even more capacity.

---

### Official Review · Reviewer_qESg · 2022-10-25

**Confidence:** 3
**Correctness:** 3
**Technical Novelty And Significance:** 3
**Empirical Novelty And Significance:** Not applicable
**Recommendation:** 6

**Clarity, Quality, Novelty And Reproducibility:**

- Clarity and quality: Many parts of the paper are unclear, as detailed above.
- Novelty: Not novel.
- Reproducibility: N/A.


**Strength And Weaknesses:**

Strengths:
* This work conducted some theoretical analysis on the fundamental limits of piecewise linearity. The study concluded the limitation of piecewise linear certification and Lipschitz-based certification under certain conditions.

Weaknesses:

* Theorem 1 talks about “piecewise-linear limited certification procedure”. But it is unclear what certification methods the paper is referring to by “piecewise-linear limited certification”. I didn’t see a concrete example in the paper for “piecewise-linear limited certification”. I know many works use convex relaxation based certification. But the limitation of convex relaxation based certification is already known as the convex relation barrier in Salman et al., 2019 which is missing in this paper. Thus I am concerned if the contribution on Theorem 1 is significant.

* Theorem 2 sounds kind of trivial to me. The existence of a Lipschitz function which can be tightly verified does not seem to be quite useful. In particular, are such Lipschitz functions nontrivial (i.e., do they correspond to any network with a good accuracy)?

* Section 3 on capacity is based on a case study only, without formal theories.

* The paper is poorly written. Many theorems are not clearly stated, especially with unclear conditions:
  - In Section 2.2.1, it is mentioned that the learner needs to be given control over the network implementation, which seems to be a condition of Theorem 2 but is missing in the Theorem.
  - In Section 2.2.2, it looks like Proposition 3 is only applicable when the networks are restricted to the hypothesis class of piecewise linear networks. This condition is also missing in Proposition 3 which simply says “Lipschitz-based certification is piecewise-linear limited.”
  - Section 2.3 is very poorly presented. The section describes some geometry  elements such as “corner”, “point”, etc., without any figure. In the current form, it is difficult for readers to understand this section. At least the writing should be combined with a figure.
  - In Section 2, “the main insights in this work stem from the simple, yet crucial observation that the points lying at a fixed Euclidean distance from a piecewise-linear decision boundary, in general, do not them- selves comprise a piecewise-linear surface.” But doesn’t certification considers points within a distance on the **input space** rather than the output space? Why do the authors consider “fixed Euclidean distance from a piecewise-linear decision boundary” (output space rather than input space).
  - In Definition 2, why does there have to be f’? Why can’t it be contained in “cert” already (the certification procedure?)

Salman, H., Yang, G., Zhang, H., Hsieh, C. J., & Zhang, P. (2019). A convex relaxation barrier to tight robustness verification of neural networks. Advances in Neural Information Processing Systems, 32.


**Summary Of The Paper:**

This is a theoretical work studying the limitations of piecewise linear functions in robustness certification. Main findings include: 1) Any piecewise linear certification is incomplete for piecewise linear networks; 2) For Lipschitz networks, when the learner can control the network, there exists some implementation such that Lipschitz-based certification is tight, but Lipschitz-based certification is piecewise-linear limited when restricted to the hy- pothesis class of piecewise linear networks. 3) Capacity can help tight certification, but using Lipschitz-based certification may need additional capacity.


**Summary Of The Review:**


The theoretical contributions are not significant enough in the current form. The writing of the paper is quite unclear. Discussion on capacity is based on a case study and lacks formal theories or solid experiments. Therefore, it seems that this paper is not ready for publication in its current form.


I suggest the authors make the writing much more clear. Theorem 1~2 needs to be supported by concrete examples and their significance needs to be improved. Section 3 on capacity needs to be supported by either formal theories or solid experiments.

---

> ### Author Response · Authors · 2022-11-07
> **Thank you for your review**
>
> Thank you for taking the time to review this work.
>
> First, there seems to be some confusion around the definitions provided, so we will clarify here and would be happy to add clarity to the writing where necessary to avoid this confusion.
>
> The fundamental misunderstanding appears to stem from here:
>
> > In Definition 2, why does there have to be f’? Why can’t it be contained in “cert” already (the certification procedure?)
>
> This is because multiple functions can implement the same decision boundary. To be complete on the hypothesis class according to our definition, the method only needs to be complete on _at least **one implementation** (f') of **every** decision boundary (F)_ in the hypothesis class. Here F is the decision boundary, and f' is a particular implementation. Hence why F is quantified universally, but f' is existential. "Completeness on a hypothesis class" is a very strong property, but not as strong as a notion of "completeness" that requires the certification must be exact on _all implementations_. We consider Definition 2 to be a useful notion, not found in other work, that is particularly relevant for discussing the strength of certifiable training methods (as opposed to post hoc certification methods).
>
> > Theorem 2 sounds kind of trivial to me. The existence of a Lipschitz function which can be tightly verified does not seem to be quite useful. In particular, are such Lipschitz functions nontrivial (i.e., do they correspond to any network with a good accuracy)?
>
> Theorem 2 is stronger than this comment suggests. Taking into account our definition of "complete over a hypothesis class," Theorem 2 says that for _any decision boundary_ that can be implemented by a Lipschitz function, there exists a particular implementation that produces the same boundary and is tightly certifiable. Indeed, this applies to nontrivial functions; e.g., a boundary that perfectly separates the data with an epsilon margin---and thus gets perfect VRA---(this obviously requires that the data are 2epsilon-separated) would be tightly certifiable.
>
> > In Section 2.2.1, it is mentioned that the learner needs to be given control over the network implementation, which seems to be a condition of Theorem 2 but is missing in the Theorem.
>
> This is because Theorem 2 is not about a learning procedure at all; it is simply about the properties of the hypothesis class and the certification method, therefore its validity does not depend on whether the "learner has control." The point about control over the decision boundary is merely to highlight the significance of Theorem 2. When learning, we always have control over the function that implements the boundary. Completeness on the hypothesis class (referenced in Theorem 2) means that the _optimal_ implementation is tightly certifiable. Thus, if certification is baked into the learning procedure, there is some hope of learning a tightly certifiable function, since this would maximize VRA. This is in contrast to post hoc certification, where we are expected to certify a given network that we had no control over. But in practice Lipschitz-based certification is not used as a post hoc method (it would be too loose), rather, state-of-the-art certified performance comes from incorporating efficient certification during training.
>
> > In Section 2.2.2, it looks like Proposition 3 is only applicable when the networks are restricted to the hypothesis class of piecewise linear networks. This condition is also missing in Proposition 3 which simply says “Lipschitz-based certification is piecewise-linear limited.”
>
> The definition of PLL captures this stipulation already. PLL means that _if the network function is piecewise linear_ then its certified frontier will be piecewise linear. Thus, Proposition 3 vacuously holds when the network is not piecewise linear.
>
> > Doesn’t certification considers points within a distance on the input space rather than the output space? Why do the authors consider “fixed Euclidean distance from a piecewise-linear decision boundary” (output space rather than input space).
>
> Perhaps this needs to be clarified in the text, but distance to the decision boundary is meant to be measured in the input space, not the output space, as the decision boundary "lives" in the input dimension. If necessary we can define decision boundary formally; informally, it is the set of points (in input space) that are directly between class labels. Naturally, certification is simply ensuring the distance to the decision boundary is sufficiently large.
>
> > Section 2.3 describes some geometry elements such as “corner”, “point”, etc., without any figure. In the current form, it is difficult for readers to understand this section. At least the writing should be combined with a figure.
>
> We agree a figure will help. We omitted this in the interest of space, but will find a way to work it back in to make the section more clear.

---

> > ### Author Response · Authors · 2022-11-07
> > **On PLL Methods and Related Work**
> >
> > It is definitely interesting to consider which methods are PLL (reviewer `gENj` also mentioned this). We will update with a survey of other PLL methods, but, of course, the primary one we consider is Lipschitz-based certification. We use the concept of PLL to make it clear that the problem with Lipschitz-based certification is not specific to its use of the Lipschitz constant, but rather, its piecewise linear certified frontier (on piecewise linear networks), which may be a problem with other methods as well.
> >
> > > I didn’t see a concrete example in the paper for “piecewise-linear limited certification”
> >
> > The concrete example that is the primary focus of in this work is Lipschitz-based certification, covered at length in Section 2.2.
> >
> > > The limitation of convex relaxation based certification is already known as the convex relation barrier in Salman et al., 2019
> >
> > This is interesting complementary work. There are some key differences, which we will add to our discussion of related work in Section 4. First, it focuses on LP-relaxed verifiers, e.g., KW, which is different from Lipschitz-based certification, which we focus on. Second, their work focuses on $\ell_\infty$, while ours focuses on $\ell_2$, using very different insights. Finally, we introduce a concept of completeness on a hypothesis class, which effectively considers the case where the learner can optimize to the certification method, while their work is on post-hoc certification. It is known that on arbitrary networks, exact certification is NP-complete. Thus, it is unsurprising that efficient methods are limited for post hoc certification (their result focuses on _how_ limited LP-relaxed verifiers are, by providing a theoretically optimal LP-relaxed verifier and empirically measuring its tightness). However, we show that tight, accurate certification is still possible using efficient certification techniques if the learner can optimize for certification (this is limited in the ways we discuss). The concept of "completeness on a hypothesis class" is new, so it is a different sort of limitation from the one discussed by Salman et al.

---

> > > ### Author Response · Authors · 2022-11-19
> > > **Some updates**
> > >
> > > We have added some updates to the paper, including a discussion of the limitations of other methods (e.g., we discuss other PLL methods besides Lipschitz-based certification) in Appendix B. The discussion we include here of convex relaxations includes a brief note on the differences between the findings of Salman et al. and our work. We will continue to make additional minor revisions regarding clarity. We hope our response and updates help clear up the misunderstandings with this work, so our contributions are more clear.

---

> > > > ### Comment · Reviewer_qESg · 2022-11-21
> > > > **Post-rebuttal update**
> > > >
> > > > Thanks to the authors for the clarification. The response has addressed most of my concerns. Overall this paper looks insightful and sound to me. Section 3 is a remaining weakness which is mentioned in this original review. Also, I think it would be helpful if there is a detailed explanation/proof for Proposition 3. It doesn't look that straightforward to me.
> > > >
> > > > I'm increasing my rating to 6.

---

### Decision · Program_Chairs · 2023-01-20

**Decision:**

Reject

**Justification For Why Not Higher Score:**

The claims in the paper are misleading and not justified by the results.

**Justification For Why Not Lower Score:**

N/A

**Metareview: Summary, Strengths And Weaknesses:**

The authors develop results showing limitations of certain classes of robustness certification procedures (those based on an analysis of global Lipschitz constants) for piecewise linear neural networks.

Strengths:
1. Shedding light on fundamental limitations of particular verification approaches is valuable since it can help focus attention on other approaches that do not suffer from these.

Weaknesses:
1. The limitations developed do not apply to SOTA methods like alpha-beta CROWN (https://github.com/Verified-Intelligence/alpha-beta-CROWN).
2. The authors do not study how the data distribution impacts this in practice, since the high density regions of the data distribution may not have significant overlap with the regions of input space where the decision boundary exhibits the properties the authors' results depend on.
3. The paper is framed too broadly and the claims are not well justified by the actual results.

Hence, I recommend rejection. The reviewers agreed though, that the paper has some interesting ideas. I would encourage the authors to submit a revision to a future venue addressing the weaknesses.

**Summary Of Ac-Reviewer Meeting:**

The discussion focused on reasons in favor of/against accepting this paper. In summary, the reviewers did not feel that there was sufficient contributions to justify acceptance, particular given the discrepancy between the way the paper is framed and the class of certification algorithms the authors' work applies to.

Pros: The paper develops fundamental limitations of a certain class of robustness certification procedures based on computing global Lipschitz bounds for piecewise linear neural networks.

Cons:

The limitations do not apply to many state of the art certification procedures based on convex relaxations + branch and bound (like https://github.com/Verified-Intelligence/alpha-beta-CROWN). However, the authors still frame the paper as being broadly applicable, which is misleading.

The authors do not study how the data distribution impacts this in practice, since the high density regions of the data distribution may not have significant overlap with the regions of input space where the decision boundary exhibits the properties the authors' results depend on.